# Inhibition of Phosphatidylinositol 3-Kinase by Pictilisib Blocks Influenza Virus Propagation in Cells and in Lungs of Infected Mice

**DOI:** 10.3390/biom11060808

**Published:** 2021-05-29

**Authors:** Stefanie Deinhardt-Emmer, Laura Jäckel, Clio Häring, Sarah Böttcher, Janine J. Wilden, Brigitte Glück, Regine Heller, Michaela Schmidtke, Mirijam Koch, Bettina Löffler, Stephan Ludwig, Christina Ehrhardt

**Affiliations:** 1Institute of Medical Microbiology, Jena University Hospital, Am Klinikum 1, D-07747 Jena, Germany; Miriam.koch@med.uni-jena.de (M.K.); bettina.loeffler@med.uni-jena.de (B.L.); 2Section of Experimental Virology, Institute of Medical Microbiology, Center for Molecular Biomedicine (CMB), Jena University Hospital, Hans-Knoell-Str. 2, D-07745 Jena, Germany; clio.haering@uni-jena.de (C.H.); sarah.boettcher@med.uni-jena.de (S.B.); glueck.brigitte@t-online.de (B.G.); michaela.schmidtke@med.uni-jena.de (M.S.); 3Institute of Virology Muenster, Centre for Molecular Biology of Inflammation (ZMBE), Westfaelische Wilhelms-University, D-48149 Muenster, Germany; laura.jaeckel@ukmuenster.de (L.J.); janine.wilden@wwu.de (J.J.W.); ludwigs@uni-muenster.de (S.L.); 4Institute of Molecular Cell Biology, Center for Molecular Biomedicine (CMB), Jena University Hospital, Hans-Knoell-Str. 2, D-07745 Jena, Germany; regine.heller@med.uni-jena.de

**Keywords:** phosphatidylinositol-3 kinase, signaling, influenza virus, Pictilisib

## Abstract

Influenza virus (IV) infections are considered to cause severe diseases of the respiratory tract. Beyond mild symptoms, the infection can lead to respiratory distress syndrome and multiple organ failure. Occurrence of resistant seasonal and pandemic strains against the currently licensed antiviral medications points to the urgent need for new and amply available anti-influenza drugs. Interestingly, the virus-supportive function of the cellular phosphatidylinositol 3-kinase (PI3K) suggests that this signaling module may be a potential target for antiviral intervention. In the sense of repurposing existing drugs for new indications, we used Pictilisib, a known PI3K inhibitor to investigate its effect on IV infection, in mono-cell-culture studies as well as in a human chip model. Our results indicate that Pictilisib is a potent inhibitor of IV propagation already at early stages of infection. In a murine model of IV pneumonia, the in vitro key findings were verified, showing reduced viral titers as well as inflammatory response in the lung after delivery of Pictilisib. Our data identified Pictilisib as a promising drug candidate for anti-IV therapies that warrant further studying. These results further led to the conclusion that the repurposing of previously approved substances represents a cost-effective and efficient way for development of novel antiviral strategies.

## 1. Introduction

Seasonal influenza virus (IV) epidemics and recurring pandemics engender respiratory illness and frequently severe disease symptoms, high mortality rates, and substantial healthcare costs [1,2]. So far, the most efficient way to protect against annual IV epidemics is vaccination [3]. Unfortunately, coverage is low, efficiency is variable, and vaccinations will not confer protection against newly emerging IV subtypes [4]. Licensed therapeutics in Germany are currently restricted to one substance class that targets the viral neuraminidase, since M2 channel blockers are no longer recommended against IV [5]. A new drug class, represented by Baloxavir marboxil, a polymerase inhibitor, is so far only approved in Japan and the United States of America [6,7]. Nonetheless, a major drawback of these antivirals is that IV can rapidly gain resistances.

Novel strategies make use of virus-supportive cellular functions to cope with viral infections [8]. IV infection leads to the activation of a variety of intracellular signaling pathways that are in part exploited by the virus to support efficient replication. The inhibition of virus-supportive cellular factors for IV therapy has a major advantage, namely, to enhance the barrier towards development of resistant virus variants [9,10].

One of these factors is the phosphatidylinositol-3-kinase (PI3K), which fulfills key-functions at different stages during IV replication with distinct outcome [11,12,13]. Already during viral attachment, the induction of PI3K promotes the internalization of the virus [13,14]. Interestingly, this PI3K-dependent mode of internalization is transferable to other virus subtypes and species [15]. Subsequently, PI3K is also involved in the regulation of the vacuolar-ATPase-dependent intracellular pH-changes of the endosome required for fusion of the viral and the endosomal membrane [16].

While early activation of the kinase is most likely due to the vRNA sensory pathway [17,18], in later stages, PI3K is the only kinase known so far, that is directly activated by binding to an influenza viral protein, the non-structural protein 1 (A/NS1), which interacts with the regulatory subunit p85 of PI3K [19,20]. Through this binding, the viral replication is supported due to prevention of pre-mature apoptosis [20,21]. Accordingly, inhibition of the PI3K pathway results in decreased viral replication, indicating the dominant virus-supportive functions of this kinase [12].

In order to find new treatment strategies for severe infections by repurposing of existing drugs for new indications, we used Pictilisib as a potent inhibitor of PI3K to target IV infection. This procedure is a cost- and timesaving process since Pictilisib has already undergone phase I and phase II clinical trials [22,23]. The first phase I study in humans was published in 2015 and showed an anti-tumor activity at doses > 100 mg with a maximum tolerated dose of 330 mg administered orally daily. Remarkably, common drug-related toxicities, such as nausea, fatigue, diarrhea, vomiting, dysgeusia, and reduced appetite were only observed at high doses and were resolved with discontinuation of Pictilisib [23]. The anti-tumor effectiveness of Pictilisib was shown for breast cancer as well as for other solid cancers [24].

The present study describes for the first time an anti-IV activity of the well-characterized PI3K inhibiting drug Pictilisib. Remarkably, its anti-IV activity was not only demonstrated in in vitro experiments, where Pictilisib effectively interferes with viral replication, but also in a murine model of IV pneumonia, where both viral titers as well as inflammatory responses were reduced in the lungs.

## 2. Materials and Methods

### 2.1. Treatment with Pictilisib

Pictilisib (GDC-0941, Selleckchem, Houston, USA) was dissolved in Dimethylsulfoxid (DMSO) and used in concentrations of 1 µM, 2.5 µM, 5 µM or 10 µM for the in vitro experiments.

For the murine model of pneumonia, C57BL/6JRj mice were treated with 150 mg kg^–1^ per os (p.o.) in 0.05% carboxymethyl cellulose (CMC, Merck KGaA, Darmstadt, Germany) and 0.02% Tween 80 (Merck KGaA, Darmstadt, Germany) as used in murine anti-cancer studies [25].

### 2.2. Cells and Viruses

For the in vitro experiments the human lung epithelial cell line A549 (American Type Culture Collection (ATCC), Wesel, Germany) and the bronchioepithelial cell line Calu-3 (ATCC, Wesel, Germany) were cultivated and grown in Dulbecco’s modified Eagle medium (DMEM; Sigma-Aldrich). The alveolar-epithelial type II cell line NCI-H441 (human papillary adenocarcinoma, ATCC, Manassas, USA) was cultivated and grown in RPMI 1640 medium (Gibco, Thermo Fisher Scientific). Both media were supplemented with 10% fetal calf serum (FCS) (Biochrom, Berlin, Germany or Merck KGaA, Darmstadt, Germany).

The Madin-Darby canine kidney (MDCK) cells have been cultured in EMEM with 10% FCS and used for virus propagation. The human influenza virus strain A/Puerto Rico/8/1934 (PR8M, H1N1), and the mouse-adapted recombinant A/Seal/Massachusetts/1/80 (SC35M, H7N7) have been propagated and passaged in MDCK cells.

For the murine model, a variant of the H1N1 influenza A virus of the pandemic 2009 (A(H1N1)pdm09) HA-G222-mpJena/5258 carrying a glycine in position 222 of the viral hemagglutinin (HA-G222-mpJena/5258) was passaged on MDCK cells. HA-G222-mpJena/5258 was obtained after three plaque-purification steps [26] from the trachea of a BALB/c mice infected with the A(H1N1)pdm09 isolate A/Jena/5258/09 [27].

### 2.3. In Vitro Infection

For the in vitro infection, A549 cells were seeded in 12-well plates (1.5 × 10^5^ cells/well) in 1 mL DMEM for the immunofluorescence experiment and in 6-well plates (5 × 10^5^ cells/well) in 2 mL DMEM for other experiments 24 h prior to infection. Calu-3 cells were seeded in 6-well plates (2 × 10^6^ cells/well) in 2 mL DMEM and NCI-H441 cells were seeded in 12-well plates (5 × 10^5^ cells/well) in 1 mL RPMI.

Cells were left untreated or washed with phosphate-buffered saline (PBS) and pre-incubated with Pictilisib or DMSO in 750 µL (12-well plate) or 1 mL (6-well plate) in DMEM/INF (DMEM supplemented with 1 mM MgCl_2_, 0.9 mM CaCl_2_, 0.2% BSA, 100 units mL^−1^ penicillin and 0.1 mg mL^−1^ streptomycin) for 30 min (2 h for immunofluorescence experiment) with the indicated concentrations.

After a washing step, cells were incubated with the indicated virus in 250 µL (12-well plate) or 500 µL (6-well plate) PBS/BA (PBS containing 0.2% bovine serum albumin (BSA), 1 mM MgCl_2_, 0.9 mM CaCl_2_, 100 units mL^−1^ penicillin, 0.1 mg mL^−1^ streptomycin). The virus solution was removed after 30 min and cells were washed with PBS before incubation with Pictilisib or DMSO in 750 µL (12-well plate) or 1 mL (6-well plate) DMEM/INF supplemented with 2 µg mL^−1^ trypsin-TPCK to the indicated time points post infection (p.i.).

### 2.4. In Vivo Infection

We used eight-week-old female specific-pathogen free (SPF) C57BL/6JRj mice (Janvier, Le Genest-Saint-Isle, France). The animal application was approved by the local ethics committee of the Thuringian State Office for Consumer Protection (Thüringer Landesamt für Verbraucherschutz, Reg.-Nr.: UKJ-018-028). The mice were maintained according to institutional guidelines in individually ventilated cages, and food and water were given ad libitum.

Pictilisib was given by oral gavage with 150 mg kg^−1^ in 0.05% carboxymethyl cellulose (CMC) and 0.02% Tween 80 or the vehicle (mock treatment). Immediately after treatment, the mice were anaesthetized by isoflurane inhalation and intranasally infected with HA-G222-mpJena/5258 [10 ^6^ plaque forming units (PFU)] in 20 μL NaCl or were mock-infected with 20 µL NaCl. The oral gavage with Pictilisib or the vehicle was given again on day 1 and 2 after infection.

During the infection the state of the animals’ health was visually checked up to two times per day. The body weight and general behavioral condition of each animal were recorded once daily after infection. Three days after virus infection, animals were sacrificed, and samples of blood and lung were subjected to further analysis.

The lobus cranialis were supplemented with EMEM according to their weight (10 µL EMEM per mg of the lung weight) and homogenized for 3 min. Subsequently, the samples were centrifuged at 3000× *g* for 3 min, and the supernatants were collected for virus titer determination and LEGENDplex ^TM^ assay.

The lobus medialis were incubated in RNA later solution (10 µL mg^−1^; RNAlater Stabilization Solution, Invitrogen, Waltham, MA, USA) over night. After removal of the RNA later solution, the tissue samples were frozen and stored at −80 °C until use.

### 2.5. Plaque Titrations

The supernatants of the indicated samples of in vitro and in vivo experiments were used to determine the number of infectious particles. For this, MDCK cells were seeded in 6-well plates until a 90% confluence and subsequently were infected with serial dilutions of the supernatants in PBS/BA for 30 min at 37 °C. After aspiration of the inoculum, cells were incubated with 2 mL EMEM/BA (EMEM with 0.2% BSA) containing 1 mM MgCl_2_, 0.9 mM CaCl_2,_ 2 µg mL^−1^ trypsin-TPCK supplemented with 0.6% agar (Oxoid, Wesel, Germany), 0.3% DEAE-Dextran (Pharmacia Biotech, Germany) and 1.5% NaHCO_3_ at 37 °C and 5% CO_2_ for 2–3 days. The viral titers were determined by standard plaque assays via visualization of the plaque-forming units (PFU) stained with neutral red.

### 2.6. Cell Growth Analysis

A549-, Calu-3- and NCl-H441 cells were seeded into 24-well plates (2.5 × 10^5^ cells/well) in 500 µL of DMEM (A549, Calu-3) or RPMI (NCI-H441) with 10% FCS 24 h prior to the experiment. The medium was removed, cells were washed with PBS, and treated with Pictilisib or DMSO at the indicated concentrations. After 4 h or after 24 h, respectively, cells were washed with PBS, and detatched with 150 µL/well Trypsin/EDTA (Sigma-Aldrich). The procedure was stopped by adding 350 µL/well medium with 10% FCS. 20 µL of suspension was supplemented with 20 µL 0.1% Trypan Blue (Invitrogen) and used for counting.

### 2.7. Western Blotting

Protein expression was determined by Western blot analysis. For this, cells were lysed on ice with RIPA buffer (25 mM Tris-HCl pH 8.0, 137 mM NaCl, 10% glycerol, 0.1% SDS, 0.5% DOC, 1% NP40, 2 mM EDTA pH 8.0, 5 µg mL^−1^ leupeptin, 5 µg mL^−1^ aprotinin, 0.2 mM pefablock, 1 mM sodium vanadate and 5 mM benzamine) for 30 min. After centrifugation, the lysates were transferred to SDS-PAGE and afterwards blotted onto a nitrocellulose membrane.

PI3K activity was monitored by detection of Akt phosphorylation with a specific phospho-Akt antibody (Ser473) (rabbit, Cell Signaling Technology, Germany). The cleaved PARP was detected by a PARP antibody (mouse, BD Transduction Laboratories, Germany). The nucleoprotein (NP), the polymerase-complex-protein 1 (PB1) and the matrix protein 1 (M1) of influenza A viruses were detected by an anti-A/NP antibody (mouse, BioRad, Germany), an anti-A/PB1 antibody (rabbit, Genetex, USA, Germany) and an anti-A/M1 antibody (mouse, BioRad, Germany). For loading control p44/42 MAPK (ERK 1/2) (137F5) (rabbit, Cell Signaling Technology, Germany) was used. The primary antibodies were diluted 1:1000 in tris-buffered saline containing 0.1% Tween 20 (TBST), supplemented with 5% BSA. A horse anti-mouse/or horse anti-rabbit IgG-HRP antibody was used as a secondary antibody at a dilution of 1:3000 in TBST. The visualization was performed with a standard chemiluminescence reaction.

### 2.8. Reporter Gene Assay

A plasmid to drive luciferase expression by a constitutively active Cytomegalovirus (CMV) promoter was transfected with Lipofectamine 2000 (Invitrogen) into A549 cells according to a protocol by Basler et al. [28]. Six hours post transfection, cells were left untreated or treated with the indicated amounts of 1 µM, 2.5 µM, 5 µM or 10 µM Pictilisib and 100 µg mL^−1^ cycloheximide (BioTrend, Germany) for 18 h. Subsequently, cells were harvested with 200 µL lysis buffer (50 mM Na-MES, pH 7.8, 50 mM Tris-HCl, pH 7.8, 1 mM dithiothreitol (DTT) and 0.2% Triton X-100).

Luciferase activity was measured and given as relative n-fold activation of nine samples as the ratio of luciferase activity relative to the DMSO control. The luciferase activity of each sample was adjusted to its protein concentration measured via Bradford protein assay.

### 2.9. Human Chip Model

In order to culture the human chip model, the chips were purchased (microfluidic ChipShop Jena, Germany) and cultivated as described previously [29,30]. For the infection procedure, the chambers of the human chip model were washed once and then incubated with 10 µM Pictilisib or DMSO for 30 min. Afterwards, the chambers were washed again and the infection with IV PR8M (0.1 MOI) was carried out by adding virus to the epithelial chamber for 30 min. Followed by a third washing step, both chambers were again treated with 10 µM Pictilisib or DMSO for 24 h. Supernatants were collected to determine viral titers by plaque assay (see Section 2.5). For the immunofluorescence microscopy, the membranes were cutted and stained with the indicated antibodies (see Section 2.10).

### 2.10. Immunofluorescence Microscopy

For the immunofluorescence microscopy of the mono-cell-culture, samples were washed with PBS twice and fixated with 300 µL of 4% formaldehyde (in PBS) (Sigma-Aldrich, Germany) for 15 min. Afterwards, cells were permeabilized for 15 min using 500 µL of 0.1% Triton X-100 (in PBS) (Roth, Germany) and unspecific binding sites were blocked for 1 h using PBS/BSA (3%). After a 2 h incubation with 50 µL of anti-A/NP antibody (mouse, BioRad, Germany) at a dilution of 1:600 in PBS/BSA, samples were washed with PBS and incubated with 50 µL of goat anti-mouse IgG (H + L) cross-adsorbed secondary antibody Alexa Fluor 658 (ThermoFisher Scientific, Germany) at a dilution of 1:1750 in PBS/BSA for 1 h. Following another washing step cells were stained with 400 µL of 4′,6′-diamidin-2-phenylindol (DAPI, Sigma-Aldrich, Germany) diluted 1:175,000 in PBS. Samples were analyzed using the fluorescence microscope Axiovert200 ™ (Zeiss, Germany).

The staining for immunofluorescence microscopy was carried out as follows: cells were washed three times with PBS and fixed by using 4% formaldehyde solution (in PBS) (Sigma-Aldrich, Darmstadt, Germany) for 15 min. The membrane was cut out, devided in half and incubated in PBS supplemented with 0.1% saponin (Sigma-Aldrich, Darmstadt, Germany) and 3% goat serum (Invitrogen) for 1 h at room temperature. The membranes were then incubated with antibodies against VE-cadherin (endothelial cells; Cell Signaling Technology, Danvers, MA, USA), E-cadherin (epithelial cells; Cell Signaling Technology, Danvers, MA, USA), and IV nucleoprotein (NP) (Bio-Rad, Hercules, CA, USA) diluted 1:200 in PBS with saponin and goat serum at 4 °C over night. The nuclei were stained with bisBenzimide H 33342 trihydrochloride (Hoechst 33342; Merck) and diluted 1:1000 in PBS/BSA. The membranes were washed with PBS and incubated with goat anti-mouse antibody (Alexa 488 coupled, Dianova, Hamburg, Germany) and goat anti-rabbit antibody (Cy-5 coupled, Dianova, Hamburg, Germany) for 30 min at room temperature. After three more washing steps the membranes were mounted on slides using flourescence mounting medium (Agilent, Santa Clara, CA, USA). Image acquisition and editing was done by using an Axio Observer.Z1 microscope (Zeiss, Jena, Germany) and the softwae Fiji V 1.52b (ImageJ).

### 2.11. qRT-PCR

For the determination of the in vitro infection, A549 cells were lysed by using the RNeasy Mini Kit (Qiagen, Hilden, Germany), according to the manufacturer’s protocol, and cDNA synthesis was performed as described earlier. For qRT-PCR, we used 2x Brilliant^®^ III SYBR Green QPCR Master Mix (Agilent Technologies, Germany) and the Roche Light Cycler^®^ 480 III (Hoffmann-La Roche, Switzerland). Primers used were GAPDH_fw 5′-GCAAATTCCATGGCACCGT-3′, GAPDH_rv 5′-GCCCCACTTGATTTGGAGG-3′, NS PR8_fw 5′-GAGGACTTGAATGGAATGATAACA-3′, NS PR8_rv 5′-GTCTCAATTCTTCAATCAATCAACCAT-3′, M1_fw 5′-TGCAAAAACATCTTCAAGTCTCTG-3′ and M1_rv 5′-AGATGAGTCTTCTAACCGAGGTCG-3′.

For in vivo experiments, lung samples were lysed by adding 350 µL RLT lysis buffer (RNeasy Mini Kit; Qiagen, Hilden, Germany) and homogenized for 3 min. After centrifugation for 3 min at 3000× *g*, the supernatant was transferred to a new reaction tube and RNA was purified using the RNeasy Mini Kit (Qiagen, Hilden, Germany), according to the manufacturer’s protocol with the QIAcube system. The RNA was diluted to equal concentrations in RNase free water, and cDNA was synthesized using the QuantiNova Reverse Transcription Kit (Qiagen, Hilden, Germany), according to the manufacturer’s protocol. Expression levels relative to GAPDH were determined by qRT-PCR analysis using the QuantiNova SYBR Green PCR Kit and the Rotor-Gene Q PCR cycler. Primers used were GAPDH_fw 5′-CAACAGCAACTCCCACTCTTC-3′, GAPDH_rv 5′- GGTCCAGGGTTTCTTACTCCTT-3′, Mx1_fw 5′-GATCCGACTTCACTTCCAGATGG-3′ and Mx1_rv 5′-CATCTCAGTGGTAGTCAACCC-3′.

### 2.12. Legendplex Assay

To determinate pro- and anti-inflammatory cytokines and chemokines, we used the LEGENDplex mouse Anti-Virus Response Panel kit (Biolegend, San Diego, CA, USA; Lot-number: B286149). For this, the supernatant of homogenized lung sections was prepared according to the manufacturer’s protocol to measure the cytokines and chemokines in pg/mL with flow cytometry (BD Accuri^TM^ C6 Plus, Heidelberg, Germany).

### 2.13. Statistical Analysis

All results were measured in at least three independent experiments. After an examination of the normal distribution of our data, parametrical methods were used for analysis. Statistical significances were evaluated by one-way-ANOVA (Tukey’s multiple comparison) and multiple *t*-tests (Holm-Sidak method). Statistical analysis was performed using Prism software (v.8; GraphPad Software, La Jolla, CA, USA).

## 3. Results

### 3.1. Pictilisib Treatment Leads to an Efficient Inhibition of Influenza Virus Replication in Mono-Cell-Culture

The efficient blockade of IV replication with experimental inhibitors of the PI3K signaling cascade has already been described previously [13]. Pictilisib differs from these compounds in that it is already in clinical use for cancer treatment [22,23,24]. In our study, we elucidated if Pictilisib may exhibit inhibitory potential on IV propagation. For this reason, we analyzed its effect on viral replication in alveolar epithelial cells (NCI-H441) (Figure 1a,b), human lung epithelial cells (A549) (Figure 1c) and bronchial epithelial cells (Calu-3) (Figure 1d). Cells were pre-incubated with the indicated concentrations of Pictilisib or solvent control (DMSO) for 30 min and infected with IV. Upon IV internalization, cells were further incubated in the presence of the indicated concentrations of Pictilisib or DMSO and progeny virus titres were determined at the indicated times.

Our results clearly show that Pictilisib is able to inhibit IV replication in vitro in a concentration-dependent manner in single-cycle (Figure 1a) and multi-cycle (Figure 1b–d) experiments independent of virus strain or cell line used.

As expected, the PI3K-Akt signaling cascade was readily activated upon IV infection, as indicated by specific phosphorylation of Akt at serine 473 (Ser473) visualized in Western blot analysis. The specific phosphorylation of Akt at serine 473 (Ser473) is required for full Akt activation and is targeted in a PI3K-dependent manner [12,31,32]. Accordingly, Pictilisib resulted in an efficient block of IV-induced Akt phosphorylation (p-Akt (S437)) (Figure 2a,b). Interestingly, the inhibitory effect of Pictilisib was independent of an additional pre-incubation of cells with the substance (Figure 2a,b). Concomitantly, we were able to show a reduction of viral protein expression upon PI3K inhibition by Pictilisib. Here, the levels of the viral proteins PB1, NP, and M1 were reduced (Figure 2c).

### 3.2. Pictilisib Treatment Does Not Affect Viability of Epithelial Cells

A major prerequisite for an antiviral agent is safety. Thus, we tested whether Pictilisib in the concentrations used would have any harming effect on cell viability. In an initial approach, we examined the interference of Pictilisib with cell growth of NCI-H441, A549 and Calu-3 cells in by cell-counting. For this analysis, various concentrations of Pictilisib were added to cells and cell growth was investigated after 4 h and after 24 h Pictilisib or DMSO treatment, respectively (Figure 3a–c). Here, no significant effect on cell growth was recognized.

While Pictilisib did not have any effects on cell growth, cellular responses such as protein synthesis still might be significantly altered in Pictilisib-treated cells. Thus, the effect of Pictilisib on cellular gene transcription and translation was assessed in a reporter gene assay using A549 cells that were transfected with a vector expressing luciferase under the control of a constitutively active CMV promoter element (Figure 3d). Here, cycloheximide was used as a positive control for the inhibition of protein synthesis. Pictilisib treatment did not alter cellular transcription or translation in this system (Figure 3d). Consistent with the previous data, Pictilisib did not induce apoptosis as evidenced by a lack of cleavage of Poly-ADP-ribose-polymerase (PARP), a prominent substrate of apoptotic caspases (Figure 3e). In summary, Pictilisib showed an unmitigated biocompatibility in our test-system.

### 3.3. An Early Stage in IV-Infection Is Inhibited by Pictilisib

It is already known that PI3K has several functions at different times of the IV replication cycle, starting with the internalization of IV particles [14]. Based on the previous results demonstrating a Pictilisib-mediated effect on viral titers (Figure 1) and viral protein expression (Figure 2c), we were prompted to analyze its effect on the level of viral mRNA synthesis. Interestingly, the data show a reduction in viral mRNA synthesis of M1 and NS1 (Figure 3f), pointing to an inhibitory effect of Pictilisib at early times of infection.

To further analyze early IV replication steps, which might be affected by Pictilisib, we performed immunofluorescence microscopy studies. Here, we monitored viral nucleoprotein (NP) localization at early times of infection in absence and presence of Pictilisib. For this, epithelial cells (A549) were infected with two different viral strains and treated with 5 µM or 10 µM Pictilisib for the indicated times. To detect an incoming virus, we applied an established immunofluorescence protocol to study the entry of IV [13,33]. The ribonucleoprotein (RNP) complexes of virus particles were stained with an antibody against the NP, the major constituent of the viral RNPs. In solvent-treated cells, NP accumulation is visible 2 h p.i. in the nucleus. In the presence of either 5 µM or 10 µM of Pictilisib, viral particles appear to be retained at the cell surface (Figure 4). Nuclear localization of NP in the presence of the inhibitor was significantly reduced, indicating a major effect on viral internalization.

### 3.4. Pictilisib Treatment Leads to an Efficient Inhibition of IV Replication in the Human Chip Model

To investigate the Pictilisib-mediated effects on IV replication in a human-specific manner in a more complex system, we used an in vitro chip model composed of cells of human origin, which was recently developed in our lab [29,30]. This model system is composed of epithelial cells (NCl-H441) and endothelial cells (HUVEC) as well as human macrophages allowing investigation of virus replication and spread across cellular barriers. IV-infected epithelial cells, treated with DMSO, reveal a productive IV infection with a visible viral NP and a disturbed epithelial layer. Interestingly, Pictilisib-treated IV-infected epithelial cells show a reduced NP accumulation at 24 h p.i. and the cell layer is dense (Figure 5a). The E-cadherin stain verifies the presence of epithelial cells and the intact barrier in DMSO-treated, uninfected cells as well as in Pictilisib-treated uninfected and IV-infected samples. In contrast, the endothelial layer that was visualized by VE-cadherin stain gets not infected independent of the treatment procedure (Figure 5b). Consistently, determination of progeny virus particles at the epithelial and endothelial site verifies ongoing virus replication in DMSO-treated samples that is blocked in the presence of Pictilisib (Figure 5c).

### 3.5. Pictilisib Blocks Virus Titers and The Pro-Inflammatory Response of IV-Infected Mice

To analyze whether the results obtained in cell culture could be transferred to the much more complex infection of a living organism, Pictilisib was applied to mice for the in vivo infection experiments. Mice were pretreated orally with Pictilisib and infected with the A/H1N1/pdm09 influenza virus variant HA-G222-mpJena/5258. Subsequently, mice were treated on day 1 and 2 after infection with Pictilisib orally with the indicated concentration.

Our results demonstrate that viral replication was significantly inhibited within the Pictilisib-treated group (Figure 6a). The virus-induced disease was indicated by the observed weight loss (Figure 6b). Here, we could not detect significant differences between the Pictilisib-treated and the mock-treated groups.

To further verify the course of disease, we analyzed different cytokines and chemokines secreted after viral infection. Host pattern-recognition receptors (PRR) recognize viral structures and initiate an effective antiviral response. This process includes the production of cytokines and chemokines which activate the inflammatory and adaptive immune response.

The release of cytokines was determined by the analysis of proteins within the lung. Here, significant lower levels for IFN-γ(Figure 6c), and CCL5 (Figure 6e) upon Pictilisib treatment were measured. CCL2 (Figure 6d), IP-10 (Figure 6f), TNF-α (Figure 6g) and IL-6 (Figure 6h) were slightly reduced. In addition, the mRNA expression of Mx1was inhibited by Pictilisib treatment within the lung of infected C57BL/6JRj mice (Figure 6i). The determination of neutrophil granulocytes within the whole blood of infected mice revealed negligible higher values for the infected mice (Figure 6j).

## 4. Discussion

Viral infections of the lung can cause severe illness with high mortality rates [34]. Since IV continuously generates new variants, vaccination is of limited effectiveness [3]. For this reason, drugs are needed that can inhibit virus replication independently of the virus strain. In this context, inhibitors of cellular signaling cascades may be very attractive candidates [8,11].

During IV infection cellular signaling factors are hijacked by the virus to ensure efficient replication and propagation. Among these, the PI3K, which is a paramount lipid kinase, is involved in key signaling events regulating intracellular metabolism. The lipid products of PI3K activate the serine/threonine kinase Akt, which is an essential factor for many cellular processes. While PI3K activity is already triggered by viral entry, the major mode of activation is mediated by binding to the viral protein NS1 protein. The fact that a viral protein is capable of inducing PI3K activity clearly shows that the virus needs to ensure PI3K activation to shape the cell for efficient propagation, including e.g., prevention of pre-mature apoptosis in late stages of viral replication [13,19,20,21].

Besides these functions in virus-infected cells, the PI3K pathway is of major importance if molecular aberrations lead to its continuous activity. This condition results in the development of numerous tumor types, for instance solid cancer including breast, colorectal and bladder, as well as various squamous cell cancers like melanoma and glioblastoma [35,36,37].

In this report, we examined the suitability of the known PI3K-inhibitor Pictilisib for the therapeutic intervention of an IV-infection. In clinical studies, Pictilisib is already applied and showed antitumor activity in breast cancer and good tolerability in humans [36]. To elucidate a potential therapeutic benefit to fight IV infection, we performed in vitro as well as in vivo assays.

We recognized that Pictilisib efficiently inhibits the propagation of IV in different epithelial cell lines in a concentration dependent manner. The results clearly indicate that Pictilisib is able to inhibit the viral replication of early and late stages of IV infection in vitro. Contemporaneously, Pictilisib shows no major toxic effects on the assayed epithelial cells. These results were verified in a more complex human chip model. Here, viral load was higher at the epithelial in comparison to the endothelial site, indicating the viral replication within the epithelial layer spreading to the endothelial site. Further, in the presence of Pictilisib, virus replication was blocked within the epithelial layer. However, also for the endothelial site there was reduced virus load observed after Pictilisib treatment, which is probably due to the reduced replication within the epithelium.

Further, we not only showed antiviral efficiency in cell culture but also in the much more complex in vivo mouse infection model. These results were in line with the in vitro results and demonstrated that Pictilisib efficiently inhibits viral replication in the mouse lung after IV infection. Although viral titers were reduced, the body weight loss was not significantly prevented. These findings suggest that Pictilisib inhibits viral replication but does not completely protect the host from virus-induced inflammatory processes. Protein concentration of IFN-γ and CCL-5 and the mRNA expression of Mx1 were reduced upon treatment with Pictilisib, while CCL-2, IP-10, TNF-α and IL-6 were slightly affected. Our data demonstrate that the antiviral immune defense is not fully impaired by the treatment.

In summary, our data indicate that repurposing of Pictilisib for anti-IV use may be a promising approach for a novel strategy of antiviral application.

## 5. Conclusions

The reutilization of compounds for the clinical application of IV infections has high potential. Since IV infections cause serious respiratory diseases, new approaches are urgently needed. The use of inhibitors of cellular signaling processes represents a suitable way to treat viral infection, while the barrier against resistance development can be enhanced.

Our study has examined whether the PI3K-inhibitor Pictilisib is able to affect IV infection. Interestingly, our results verify Pictilisib as a potential substance for an antiviral therapy, since the viral replication is significantly inhibited in vitro and in vivo. Further studies to investigate the effect are required and might contribute to a more effective therapy against IV.

## Figures and Tables

**Figure 1 biomolecules-11-00808-f001:**
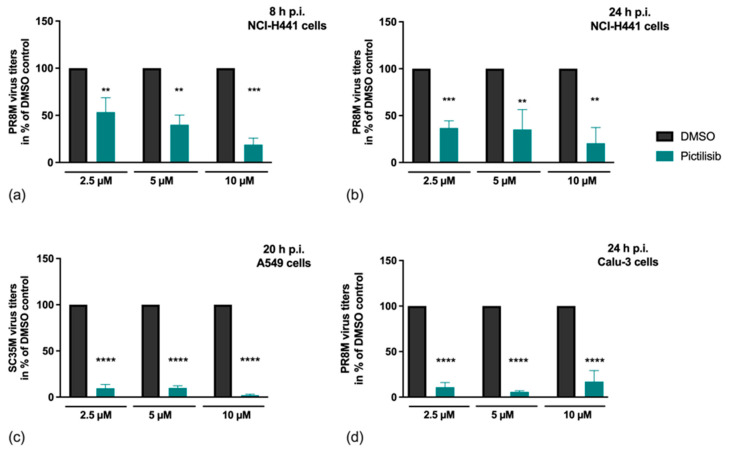
Efficient inhibition of viral replication upon Pictilisib treatment. (**a**–**d**) NCl-H441 cells (**a**,**b**), A549 cells (**c**) or Calu-3 cells (**d**) were pre-incubated with the indicated concentrations of Pictilisib or solvent control (DMSO) for 30 min prior to infection. Afterwards, viral infection was performed with PR8M (H1N1) for 8 h (MOI 0.5) (**a**), 24 h (MOI 0.01) (**b**), (MOI 0.5) (**d**), or SC35M (H7N7; MOI 0.001) for 20 h (**c**) in the presence of the indicated concentrations of Pictilisib or DMSO. Data represent means + SD of three independent experiments including two (**a**,**b**,**d**) or four (**c**) biological samples. DMSO-treated cells were arbitrarily set as 100%. Statistical analysis was performed by one-way ANOVA followed by multiple comparisons test. (**** *p* < 0.0001, *** *p* < 0.001, ** *p* < 0.01).

**Figure 2 biomolecules-11-00808-f002:**
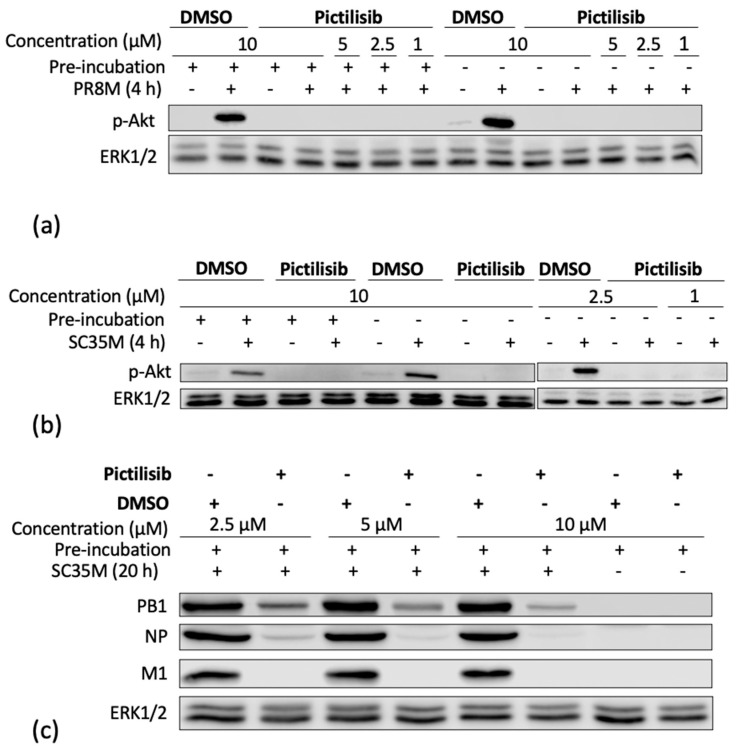
Pictilisib efficiently inhibits IV-induced Akt activation and affects early stages of viral infection. A549 cells were left untreated or pre-incubated with the indicated concentrations of Pictilisib or solvent control (DMSO) for 30 min prior to infection with influenza PR8M (H1N1) (MOI 5) (**a**) and the recombinant SC35M (H7N7) (**b**: MOI 5; **c**: MOI 0.001). After 30 min of infection, viral inoculum was removed, and cells were supplemented with medium containing Pictilisib or DMSO. After the indicated times, cell-lysates were prepared and applied to protein detection by Western blot analysis. Activation of PI3K-mediated signaling was monitored by phosphorylated Akt (Ser473). Equal protein loading of the kinase was verified by detection of ERK1/2. Ongoing viral replication was demonstrated by accumulation of the viral proteins in two (NP, M1) or three (PB1) independent experiments.

**Figure 3 biomolecules-11-00808-f003:**
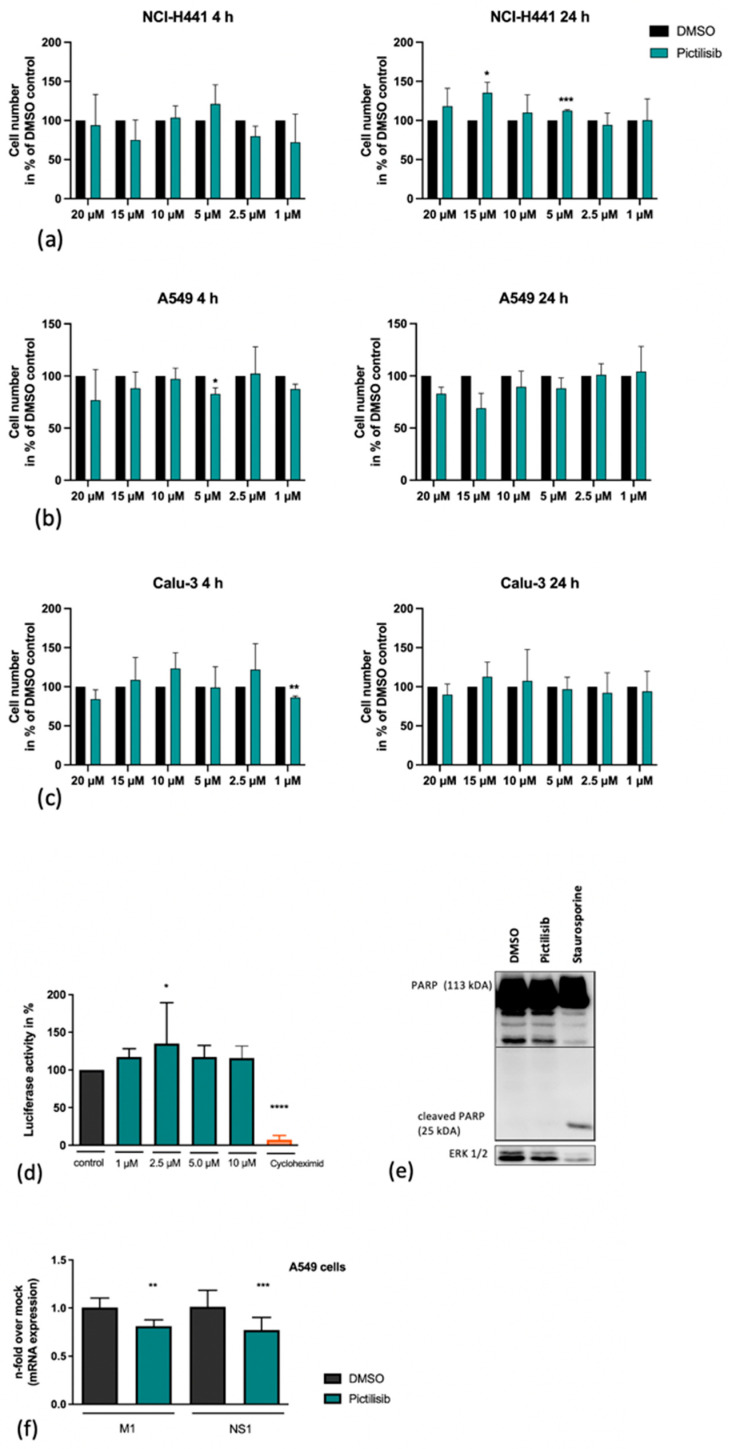
Pictilisib treatment does not affect cell viability. NCl-H441 (**a**), A549 cells (**b**) and Calu-3 cells (**c**) were treated with increasing amounts of Pictilisib (1–20 µM) or DMSO in 500 µL of medium with 10% FCS. After the indicated times cells were washed with PBS and detatched with 150 µL/well Trypsin/EDTA. Subsequently, the procedure was stopped by adding 350 µL/well medium with 10% FCS. 20 µL of suspension was supplemented with 20 µL 0.1% Trypan Blue and used for counting. (**d**) A549 cell were transfected with a constitutive active CMV promoter luciferase plasmid. Eight hours post transfection cells were left untreated or treated with the indicated amounts of Pictilisib or cycloheximide (100 µg mL^−1^) respectively for 18 h. (**d**–**f**) Staurosporine (1 µM) or cycloheximide (100 µg mL^−1^)-treated cells were used as positive controls. The cleavage of PARP (113 kDa) by Pictilisib was proven by Western blot analysis (**e**). For this, cells were lysed on ice with RIPA buffer and transferred to SDS-PAGE and afterwards blotted onto a nitrocellulose membrane. The cleaved PARP was visualized by use of a PARP antibody (**e**). A549 cells (**f**) were left untreated or pre-incubated with Pictilisib (10 µM) or DMSO for 30 min prior to infection with influenza PR8M (H1N1) (MOI 5). After 30 min of infection, viral inoculum was removed, and cells were supplemented with medium containing Pictilisib or DMSO for 8 h. The mRNA expression of M1 and NS1 was blocked significantly by Pictilisib (**f**). Data represent means + SD of three independent experiments including four biological samples. DMSO-treated cells were arbitrarily set as 100%. Statistical analysis was performed by one-way ANOVA followed by multiple comparisons test. (**** *p* < 0.0001, *** *p* < 0.001, ** *p* < 0.01, * *p* < 0.05).

**Figure 4 biomolecules-11-00808-f004:**
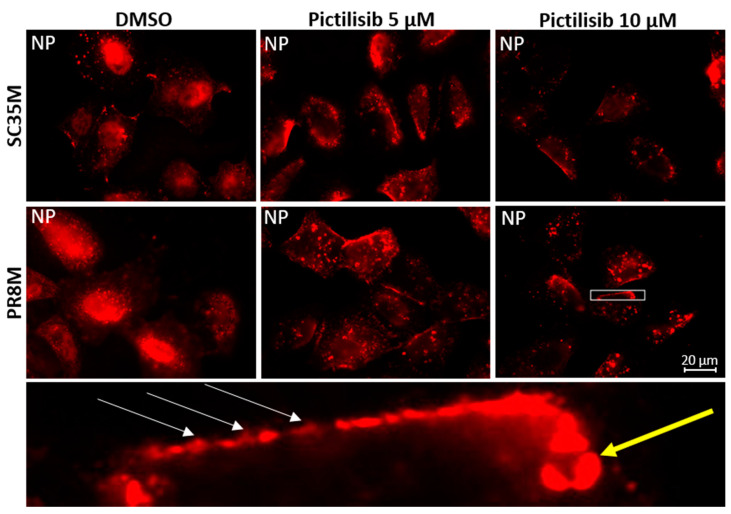
Viral internalization is impaired upon Pictilisib treatment. A549 cells were pre-incubated for 2 h with DMSO or Pictilisib (5 µM or 10 µM) in DMEM and infected with recombinant SC35M or PR8M (MOI 200). Staining was performed with primary antibody against NP. The figure is representative for three independent experiments. White arrows show virus particles at the cell border; the yellow arrow shows a large accumulation of NP in the cytoplasm. Acquired with Axiovert200M, scale bar represents 20 µm

**Figure 5 biomolecules-11-00808-f005:**
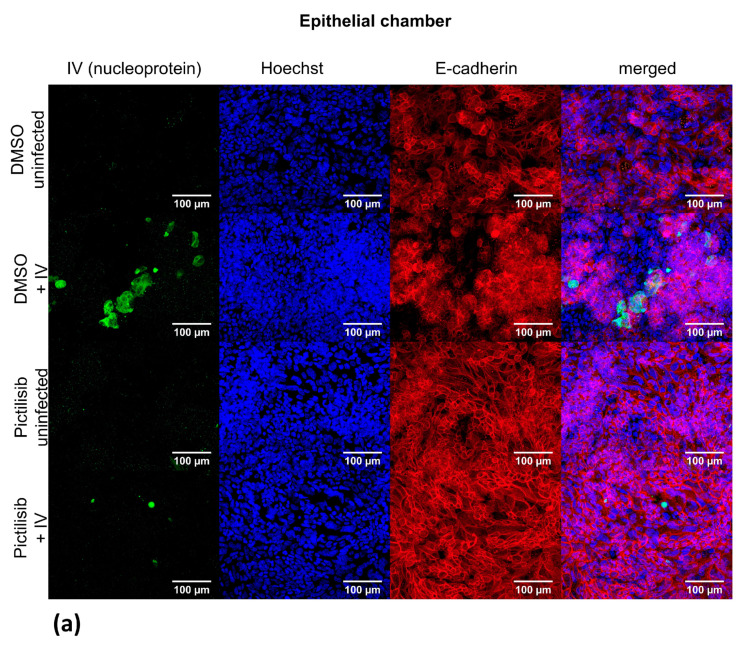
Pictilisib inhibits viral replication in the human chip model. The epithelial and endothelial sites of the human chip model were pre-incubated with 10 µM Pictilisib or solvent control (DMSO) for 30 min prior to infection. PR8M infection (MOI 0.1) was then performed for 30 min on the epithelial site of the model. The chip was incubated with Pictilisib (10 µM) or DMSO to 24 h p.i. (**a**,**b**) The cell layers of the epithelial (**a**) and endothelial chamber (**b**) were stained using primary antibodies against IV nucleoprotein (a,b), E-cadherin (**a**) and VE-cadherin (**b**), Nuclei were stained with Hoechst (**a**,**b**). (**c**) Viral titers were determined by standard plaque assay. The solvent control was arbitrarily set to 100%. Data represents the mean + SD of eight independent experiments. Statistical significance was determined by multiple *t*-tests (**** *p* < 0.0001).

**Figure 6 biomolecules-11-00808-f006:**
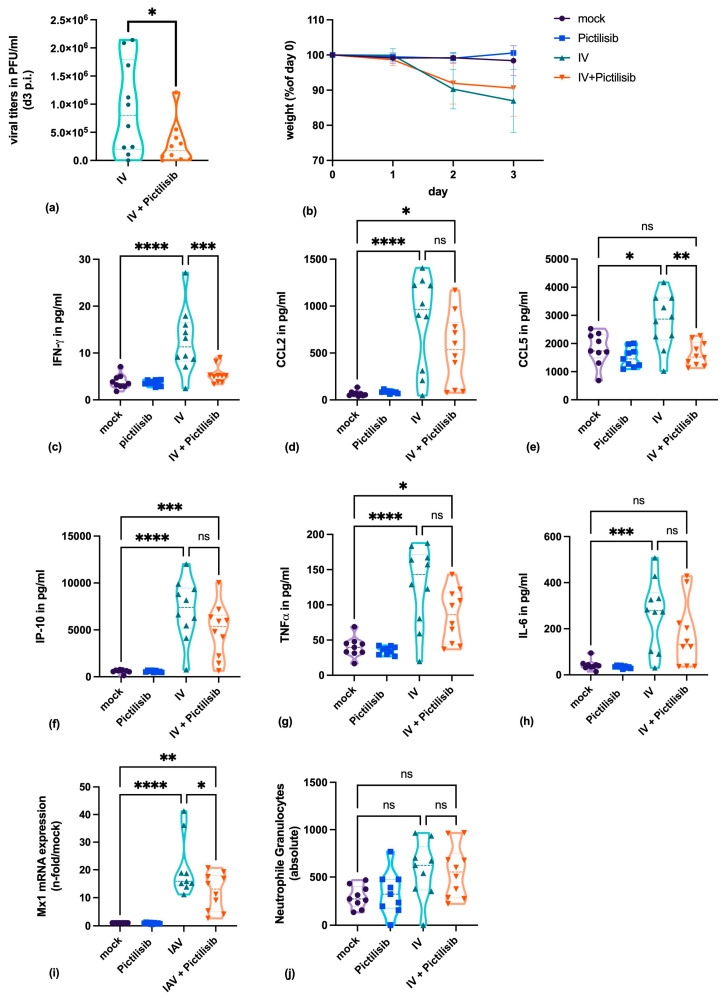
Pictilisib reduces virus titers and the pro-inflammatory response of IV-infected mice. (**a**–**j**) C57BL/6JRj mice were treated with 150 mg kg^−1^ Pictilisib on day 0 and infected with IV variant HA-G222-mpJena/5258 followed by further Pictilisib treatment on day 1 and 2 after infection. Three days post infection (p.i.) mice were sacrificed to subject blood samples and lung explants. Viral titers (**a**), body weight of mice (**b**), and the secreted pro-inflammatory cytokines and chemokines (**c**–**h**; IFN-γ, CCL2, CCL5, IP-10, TNF-α, IL-6), Mx1 mRNA expression (**i**) and number of neutrophil granulocytes (**j**) are depicted for achievable interpretation of Pictilisib treatment in mice. Statistical analysis was performed by one-way ANOVA followed by multiple comparisons test. (ns = not significant, **** *p* < 0.0001, *** *p* < 0.001, ** *p* < 0.01, * *p* < 0.05).

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
