# Peer review of "Inhibition of Phosphatidylinositol 3-Kinase by Pictilisib Blocks Influenza Virus Propagation in Cells and in Lungs of Infected Mice"

_biomolecules, 2021, doi:10.3390/biom11060808_

Round 1

Reviewer 1 Report

This reviewer understands that authors could not perform additional mouse work due to animal restrictions and appreciates the efforts of producing additional data in a human-based cell-culture model system. 

Reviewer 2 Report

In the manuscript entitled "Inhibition of phosphatidylinositol 3-kinase by Pictilisib blocks influenza virus propagation in cells and in lungs of infected mice," the authors have appropriately satisfied all of my comments and concerns.

This manuscript is a resubmission of an earlier submission. The following is a list of the peer review reports and author responses from that submission.

Round 1

Reviewer 1 Report

In this paper, Deinhardt-Emmer et al. investigated the preclinical efficacy of the anti-PI3K treatment Pictilisib against influenza virus. They found that Pictilisib reduced viral replication in both cell cultures and an animal model.

In my opinion, the results shown raise some concerns about safety of Pictilisib. Moreover, there are a few minor points that I would like to see addressed.

  1. In the Results (section 3.2), the Authors state that “Pictilisib is well tolerated with no obvious toxic effects”. However, they show that 24h treatment with Pictilisib reduced cell viability (Figures 3 a,b). Thus it seems that Pictilisib may have important toxic effects. The Authors should correct their statement, and further investigate the toxicity of Pictilisib.
  2. On this issue, in the Introduction the Authors state that Pictilisib showed good tolerability in human studies (Ref. 24). While Pictilisib tolerability can be judged good for an anti-tumor drug, I am not sure its tolerability can be judged as good for an anti-flu drug. Actually, Ref. 24 shows that Pictilisib induced 46 adverse events in 60 patients. The Authors should discuss about it.
  3. In the Statistical Methods, the Authors state that they used non-parametrical methods. However, they also state that statistical significances were evaluated by one-way ANOVA, which is a parametrical test. Did the Authors use parametrical or non-parametrical methods? Were the data normally distributed?
  4. Basal Akt phosphorylation at serine 473 is only an indirect marker of PI3K activity, since it can be influenced also by other proteins, such as mTor and PTEN.

Reviewer 2 Report

The study by Deinhardt-Emmer and colleagues investigates the potential therapeutic benefit of Pictilisib, a pan-PI3K inhibitor, against the lung disease induced by Influenza virus (IV). Authors exploit both in vitro and in vivo experimental models that are lung epithelial cell lines and IV-infected mice, respectively. They claim that, by preventing IV-induced PI3K/Akt signaling activation, Pictilisib inhibits viral replication in vitro. Furthermore, they show that in vivo treatment with Pictilisib reduces virus titer and pro-inflammatory response in infected mice.

Although the ability of PI3K inhibitors to inhibit virus replication is well established (PMID 22345452; 25751541), Pictilisib has been approved very recently by FDA for the treatment of advanced breast cancer and its potential utility in the treatment of airway disease has not been addressed yet. Hence, the study deals with a potential interesting and very actual topic considering the current COVID-19 pandemic. However, my enthusiasm is limited by major methodological and conceptual concerns.

  • In figure 1, authors investigate the effect of Pictilisib in viral replication by using two different cell lines, namely lung adenocarcinoma cells (A549) and alveolar-epithelial type II cells (NCI-H441). The two lines are infected with two different viral strains, H7Nt IV and H1N1, respectively. The choice of infecting the cells with two different strains complicates the interpretation of results. There is indeed a clear difference in virus replication upon Pictilisib treatment in the two experimental models, but it is unclear whether this is due to the cell type or the viral strain. Authors should clarify the rationale behind the choice of cell types, MOI and timepoints.
  • In general, the use of an adenocarcinoma cell line like A549 for assessing the therapeutic potential of Pictilisib in non-tumoral pathologies is debatable.
  • In Figure 2, authors analyze Akt phosphorylation as a readout of PI3K activation after IV infection, and treatment with Pictilisib. The expression levels of total Akt and of standard housekeeper genes (like Gapdh) should be provided instead of Erk1/2 expression. Relative quantification and statistics of P-Akt/Akt ratio should also be provided.
  • Authors claim that Pictilisib does not affect cell proliferation or survival, but in Figure 3 a significant reduction in proliferation at 24 hours is observed. Furthermore, to conclusively demonstrate that the compound is not toxic for lung cells, survival should be assessed by using doses that are up to 100-fold higher than the active one. Furthermore, in tumor-derived cells and in the presence of serum, proliferation and survival can be hardly distinguished. MTT assays should be performed under starvation.
  • In figure 3d, PARP signal is saturated. Please provide better representative picture, along with standard housekeeper genes. Quantification of cleaved over total PARP and statistical analysis should be performed.
  • In figure 5, the use of body weight monitoring for assessing the severity of disease in vivo is debatable. In Figure 5b, body weight is not significantly different between mock and infected mice. Additional endpoint of disease severity should be evaluated (e.g. lung resistance/capacity).
  • In figure 4f, authors show that Pictilisib does not affect IL-6 production in vivo. This contrasts with previous work showing that the PI3K inhibitor LY294002 suppresses IL 6 production upon PR8-infection [PMID: 24971535]. How do the authors explain this discrepancy? Is it a matter of isoform specificity of the different PI3K inhibitors?
  • Although Pictilisib is considered a pan-PI3K inhibitor, it preferentially targets α and δ isoforms. This opens the possibility that, by inhibiting leukocyte-restricted PI3K δ, Pictilisib directly inhibits lung inflammation. Is this the explanation for the reduction of IFN-g and CCL5 in IV-infected Pictilisib-treated mice? Immune cell counts in the BALF of these mice should be provided.
  • In paragraph 3.3, authors claim an effect of Pictilisib on early stages of infection, but it is not clear whether they refer to replication (line 302) or internalization (line 311).
  • The discussion should be significantly improved because it stands as a repetition of major findings, while it should discuss major results in relation to previously published literature. Furthermore, when referring to previous work on the role of PI3K in viral replication, authors primarily refer to their own previous work and should instead cite other relevant literature (e.g. 22345452; 25751541).

Reviewer 3 Report

In the manuscript entitled “Inhibition of phosphatidylinositol 3-kinase by Pictilisib blocks influenza virus propagation in cells and in lungs of infected mice,” I appreciate the novelty and importance of your study as influenza virus has no border limits throughout the world.

What method for delivering the oral treatment was used?

Below are suggestions that I made that may help improve the readability and impact of your manuscript:

Line 29: May I suggest using “therapies that warrant further studying, to better complete your sentence?

Line 30: Suggest inserting “a” between represents and cost-effective; also suggest substituting the word “efficient” for the word “fast”

Line 38: Suggest substituting the word “will” instead of “would”

Line 40: Add an “s” on the word blocker.

Line 63: Suggest using “Pictilisib has already undergone” phase ………..

Line 70: Add an “s” to the word lung.

Line 76: Misspelling und should be and

Line 87: Suggest replacing the words have been with the word were

Line 91: Mistype at beginning of second sentence, HA-D222-mpJena/5258 should be HA-G222-mpJena/5258

Line 112: Suggest the word “approved” in place of “proven”

Line 115: Add a comma after the word cages.

Line 121: Suggest the word animals’ in place of the word animal

Line 122: Suggest the word behavioral in place of behavior; Suggest adding “of each animal” between the words condition and were

Line 123: Add a comma after the word infection.  Add a comma after the word sacrifice.

Line 127: Add a comma after 3 min.

Line 130: In the middle of the second sentence, add a comma after the word solution.

Line 135: Add the word “were” after the term subsequently.

Line 136: Add a comma after the word inoculum.

Line 140: Suggest substituting “stained with neutral red” after (PFU) and delete “by the staining with neutral red.”

Line 145: Suggest saying “ After 4 h and again after 24 h, thiazolyl blue…..

Line 146: Misspelling, solved should be dissolved; add commas before dissolved and was.

Line 155: Add a comma after centrifugation.

Line 171: Add a comma after transfection.

Line 173: Add a comma after subsequently.

Line 179: Add a comma after microscopy.

Line 180: Add a comma after afterwards.

Line 191: Add a comma before  the word according, and after the word protocol.

Line 192: Add a comma before we.

Line 201: Add a comma before the word according.

Line 202: Add a comma after the word water.

Line 203: Add a comma before the word according.

Lines 210-212: The sentence is long and needs adjusting for better clarity.

Line 222: Add a comma after the word study.

Line 227: Add a comma after the word internalization.

Lines 227-228: For a better read, perhaps you could say: …..cells were further incubated in the previously indicated concentrations of Pictilisib or DMSO for the time periods earlier described.

Line 253: Add a comma after the word infection.

Line 255: Add a comma after the word times.

Line 257: Suggest deleting the word And and start the sentence with Ongoing

Line 262: Add a comma after the word approach.

Line 264: Suggest adding the word analysis after the word this

Line 265: Suggest adding the words “again after 24 h” after the word and.

Line 265: Add a comma after the word assay.

Line 272: Insert the word again before the word after.  Add a comma after 24 h; add a comma after (MTT, 25 ul), and add a comma after (5 mg MTT ml-1).

Line 272: Misspelling? Solved should be dissolved?

Line 298: Add a comma after (Figure 2 c).

Line 303: Add a comma after the word Here.

Line 308: Suggest using the comma after the word cells instead of after the term newly.

Line 309: Suggest deleting the word already

Line 309: Suggest spelling what is indicated by “p.i.”

Line 309: Add a comma after the word Pictilisib.

Line 317: Suggest using a semicolon after the term border instead of a comma.

Line 333: Suggest spelling what the shortcut “p.i.” indicates.

Line 374: Suggest substituting the word “assayed” for “used”

Line 376: Suggest using “with” instead of “to”

Round 2

Reviewer 2 Report

Authors did not satisfactorily address most of the points I have raised.

In response to my requests, authors simply provide argumentations while no experimental data or details are added. If in some cases the arguments provided might be convincing, in other cases they are absolutely not acceptable. For instance, authors state that “the endpoint of disease severity was not in the scope of this study”. In the absence of more appropriate endpoints and data, this in vivo experiment is not informative and the biological/clinical relevance of Pictilisib treatment in influenza virus infection cannot be determined.

As such, major methodological and conceptual concerns remain.